# ITERATIVE AMORTIZED POLICY OPTIMIZATION

## ABSTRACT

Policy networks are a central feature of deep reinforcement learning (RL) algorithms for continuous control, enabling the estimation and sampling of high-value actions. From the variational inference perspective on RL, policy networks, when employed with entropy or KL regularization, are a form of amortized optimization, optimizing network parameters rather than the policy distributions directly. However, this direct amortized mapping can empirically yield suboptimal policy estimates and limited exploration. Given this perspective, we consider the more flexible class of iterative amortized optimizers. We demonstrate that the resulting technique, iterative amortized policy optimization, yields performance improvements over direct amortization methods on benchmark continuous control tasks.

## 1 INTRODUCTION

Reinforcement learning (RL) algorithms involve policy evaluation and policy optimization (Sutton & Barto, 2018). Given a policy, one can estimate the value for each state or state-action pair following that policy, and given a value estimate, one can improve the policy to maximize the value. This latter procedure, policy optimization, can be challenging in continuous control due to instability and poor asymptotic performance. In deep RL, where policies over continuous actions are often parameterized by deep networks, such issues are typically tackled using regularization from previous policies (Schulman et al., 2015; 2017) or by maximizing policy entropy (Mnih et al., 2016; Fox et al., 2016). These techniques can be interpreted as variational inference (Levine, 2018), using optimization to infer a policy that yields high expected return while satisfying prior policy constraints. This smooths the optimization landscape, improving stability and performance (Ahmed et al., 2019).

However, one subtlety arises: when used with entropy or KL regularization, policy networks perform *amortized* optimization (Gershman & Goodman, 2014). That is, rather than optimizing the action distribution, e.g. mean and variance, many deep RL algorithms, such as soft actor-critic (SAC) (Haarnoja et al., 2018b;c), instead optimize a network to output these parameters, *learning* to optimize the policy. Typically, this is implemented as a direct mapping from states to action distribution parameters. While *direct* amortization schemes have improved the efficiency of variational inference as encoder networks (Kingma & Welling, 2014; Rezende et al., 2014; Mnih & Gregor, 2014), they are also suboptimal (Cremer et al., 2018; Kim et al., 2018; Marino et al., 2018b). This suboptimality is referred to as the amortization gap (Cremer et al., 2018), translating into a gap in the RL objective. Likewise, direct amortization is typically restricted to a single estimate of the distribution, limiting the ability to sample diverse solutions. In RL, this translates into a deficiency in exploration.

Inspired by techniques and improvements from variational inference, we investigate *iterative* amortized policy optimization. Iterative amortization (Marino et al., 2018b) uses gradients or errors to iteratively update the parameters of a distribution. Unlike direct amortization, which receives gradients only *after* outputting the distribution, iterative amortization uses these gradients *online*, thereby learning to perform iterative optimization. In generative modeling settings, iterative amortization tends to empirically outperform direct amortization (Marino et al., 2018b;a), with the added benefit of finding multiple modes of the optimization landscape (Greff et al., 2019).

Using MuJoCo environments (Todorov et al., 2012) from OpenAI gym (Brockman et al., 2016), we demonstrate performance improvements of iterative amortized policy optimization over direct amortization in model-free and model-based settings. We analyze various aspects of policy optimization, including iterative policy refinement, adaptive computation, and zero-shot optimizer transfer. Identifying policy networks as a form of amortization clarifies suboptimal aspects of direct approaches

to policy optimization. Iterative amortization, by harnessing gradient-based feedback during policy optimization, offers an effective and principled improvement.

## 2 BACKGROUND

### 2.1 PRELIMINARIES

We consider Markov decision processes (MDPs), where $\mathbf{s}_t \in \mathcal{S}$ and $\mathbf{a}_t \in \mathcal{A}$ are the state and action at time $t$, resulting in reward $r_t = r(\mathbf{s}_t, \mathbf{a}_t)$. Environment state transitions are given by $\mathbf{s}_{t+1} \sim p_{\text{env}}(\mathbf{s}_{t+1}|\mathbf{s}_t, \mathbf{a}_t)$, and the agent is defined by a parametric distribution, $p_\theta(\mathbf{a}_t|\mathbf{s}_t)$, with parameters $\theta$. The discounted sum of rewards is denoted as $\mathcal{R}(\tau) = \sum_t \gamma^t r_t$, where $\gamma \in (0, 1]$ is the discount factor, and $\tau = (\mathbf{s}_1, \mathbf{a}_1, \dots)$ is a trajectory. The distribution over trajectories is:

$$p(\tau) = \rho(\mathbf{s}_1) \prod_{t=1}^{T} p_{\text{env}}(\mathbf{s}_{t+1}|\mathbf{s}_t, \mathbf{a}_t) p_\theta(\mathbf{a}_t|\mathbf{s}_t), \tag{1}$$

where the initial state is drawn from the distribution $\rho(\mathbf{s}_1)$. The standard RL objective consists of maximizing the expected discounted return, $\mathbb{E}_{p(\tau)}[\mathcal{R}(\tau)]$. For convenience of presentation, we use the undiscounted setting ($\gamma = 1$), though the formulation can be applied with any valid $\gamma$.

### 2.2 KL-REGULARIZED REINFORCEMENT LEARNING

Various works have formulated RL, planning, and control problems in terms of probabilistic inference (Dayan & Hinton, 1997; Attias, 2003; Toussaint & Storkey, 2006; Todorov, 2008; Botvinick & Toussaint, 2012; Levine, 2018). These approaches consider the agent-environment interaction as a graphical model, then convert reward maximization into maximum marginal likelihood estimation, learning and inferring a policy that results in maximal reward. This conversion is accomplished by introducing one or more binary observed variables (Cooper, 1988), denoted as $\mathcal{O}$, with

$$p(\mathcal{O} = 1|\tau) \propto \exp\left(\mathcal{R}(\tau)/\alpha\right),$$

where $\alpha$ is a temperature hyper-parameter. These new variables are often referred to as "optimality" variables (Levine, 2018). We would like to infer latent variables, $\tau$, and learn parameters, $\theta$, that yield the maximum log-likelihood of optimality, i.e. $\log p(\mathcal{O} = 1)$. Evaluating this likelihood requires marginalizing the joint distribution, $p(\mathcal{O} = 1) = \int p(\tau, \mathcal{O} = 1) d\tau$. This involves averaging over all trajectories, which is intractable in high-dimensional spaces. Instead, we can use variational inference to lower bound this objective, introducing a structured approximate posterior distribution:

$$\pi(\tau|\mathcal{O}) = \prod_{t=1}^{T} p_{\text{env}}(\mathbf{s}_{t+1}|\mathbf{s}_t, \mathbf{a}_t) \pi(\mathbf{a}_t|\mathbf{s}_t, \mathcal{O}). \tag{2}$$

This provides the following lower bound on the objective, $\log p(\mathcal{O} = 1)$:

$$\log \int p(\mathcal{O} = 1|\tau) p(\tau) d\tau \geq \int \pi(\tau|\mathcal{O}) \left[ \log p(\mathcal{O} = 1|\tau) + \log \frac{p(\tau)}{\pi(\tau|\mathcal{O})} \right] d\tau \tag{3}$$

$$= \mathbb{E}_\pi[\mathcal{R}(\tau)/\alpha] - D_{\text{KL}}(\pi(\tau|\mathcal{O}) \| p(\tau)). \tag{4}$$

Equivalently, we can multiply by $\alpha$, defining the variational RL objective as:

$$\mathcal{J}(\pi, \theta) \equiv \mathbb{E}_\pi[\mathcal{R}(\tau)] - \alpha D_{\text{KL}}(\pi(\tau|\mathcal{O}) \| p(\tau)) \tag{5}$$

This objective consists of the expected return (i.e., the standard RL objective) and a KL divergence between $\pi(\tau|\mathcal{O})$ and $p(\tau)$. In terms of states and actions, this objective is written as:

$$\mathcal{J}(\pi, \theta) = \mathbb{E}_{\substack{\mathbf{s}_t, r_t \sim p_{\text{env}} \\ \mathbf{a}_t \sim \pi}} \left[ \sum_{t=1}^{T} r_t - \alpha \log \frac{\pi(\mathbf{a}_t|\mathbf{s}_t, \mathcal{O})}{p_\theta(\mathbf{a}_t|\mathbf{s}_t)} \right]. \tag{6}$$

At a given timestep, $t$, one can optimize this objective by estimating the future terms in the summation using a "soft" action-value ($Q_\pi$) network (Haarnoja et al., 2017) or model (Piché et al., 2019). For instance, sampling $\mathbf{s}_t \sim p_{\text{env}}$, slightly abusing notation, we can write the objective at time $t$ as:

$$\mathcal{J}(\pi, \theta) = \mathbb{E}_\pi[Q_\pi(\mathbf{s}_t, \mathbf{a}_t)] - \alpha D_{\text{KL}}(\pi(\mathbf{a}_t|\mathbf{s}_t, \mathcal{O}) \| p_\theta(\mathbf{a}_t|\mathbf{s}_t)). \tag{7}$$

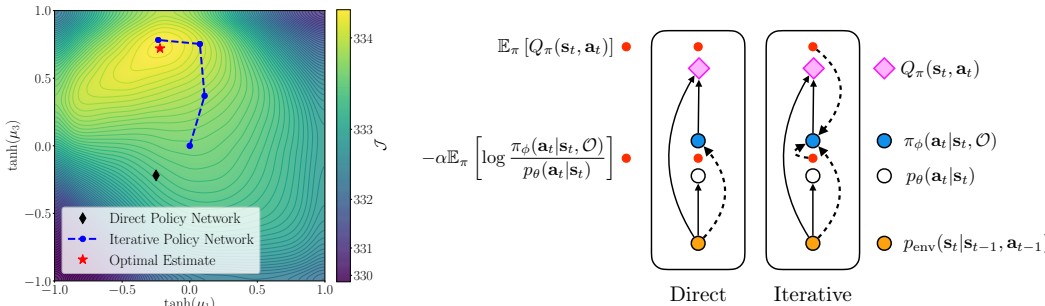

Figure 1: **Amortization**. **Left**: Optimization over two dimension of the policy mean, $\mu_1$ and $\mu_3$, for a particular state. A direct amortized policy network outputs a suboptimal estimate, yielding an *amortization gap* in performance. An iterative amortized policy network finds an improved estimate. **Right**: Diagrams of direct and iterative amortization. Larger circles denote distributions, and smaller red circles denote terms in the objective, $\mathcal{J}$ (Eq. 7). Dashed arrows denote amortization. Iterative amortization uses gradient feedback during optimization, while direct amortization does not.

Policy optimization in the KL-regularized setting corresponds to maximizing $\mathcal{J}$ w.r.t. $\pi$. We often consider parametric policies, in which $\pi$ is defined by distribution parameters, $\boldsymbol{\lambda}$, e.g. Gaussian mean, $\boldsymbol{\mu}$, and variance, $\boldsymbol{\sigma}^2$. In this case, policy optimization corresponds to maximizing:

$$\boldsymbol{\lambda} \leftarrow \arg\max_{\boldsymbol{\lambda}} \mathcal{J}(\pi, \theta). \tag{8}$$

Optionally, we can then also learn the policy prior parameters, $\theta$ (Abdolmaleki et al., 2018).

### 2.3 KL-REGULARIZED POLICY NETWORKS PERFORM DIRECT AMORTIZATION

Policy-based approaches to RL typically do not directly optimize the action distribution parameters, e.g. through gradient-based optimization. Instead, the action distribution parameters are output by a function approximator (deep network), $f_\phi$, which is trained using deterministic (Silver et al., 2014; Lillicrap et al., 2016) or stochastic gradients (Williams, 1992; Heess et al., 2015). When combined with entropy or KL regularization, this policy network is a form of *amortized* optimization (Gershman & Goodman, 2014), learning to estimate policies. Again, denoting the action distribution parameters, e.g. mean and variance, as $\boldsymbol{\lambda}$, for a given state, $\mathbf{s}$, we can express this direct mapping as

$$\boldsymbol{\lambda} \leftarrow f_\phi(\mathbf{s}), \qquad \text{(direct amortization)} \tag{9}$$

and we denote the corresponding policy as $\pi_\phi(\mathbf{a}|\mathbf{s}, \mathcal{O}; \boldsymbol{\lambda})$. Thus, $f_\phi$ attempts to *learn* to optimize Eq. 8. This setup is shown in Figure 1 (Right). Without entropy or KL regularization, i.e. $\pi_\phi(\mathbf{a}|\mathbf{s}) = p_\theta(\mathbf{a}|\mathbf{s})$, we can instead interpret the network as directly integrating the LHS of Eq. 3, which is less efficient and more challenging. Adding regularization smooths the optimization landscape, resulting in more stable improvement and higher asymptotic performance (Ahmed et al., 2019).

Viewing policy networks as a form of amortized variational optimizer (inference model) (Eq. 9) allows us to see that they are similar to encoder networks in variational autoencoders (VAEs) (Kingma & Welling, 2014; Rezende et al., 2014). This raises the following question: *are policy networks providing fully-optimized policy objectives?* In VAEs, it is empirically observed that amortization results in suboptimal approximate posterior estimates, with the resulting gap in the variational bound referred to as the *amortization gap* (Cremer et al., 2018). In the RL setting, this means that an amortized policy, $\pi_\phi$, results in worse performance than the optimal policy within the parametric policy class, which we denote as $\widehat{\pi}$. Thus, the amortization gap is the gap in following inequality:

$$\mathcal{J}(\pi_\phi, \theta) \leq \mathcal{J}(\widehat{\pi}, \theta).$$

Because $\mathcal{J}$ is a variational bound on the RL objective, i.e. expected return, a looser bound, due to amortization, prevents an agent from more completely optimizing this objective.

To visualize the RL amortization gap, in Figure 1 (Left), we display the optimization surface, $\mathcal{J}$, for two dimensions of the policy mean at a particular state in the MuJoCo environment `Hopper-v2`.

**Algorithm 1** Direct Amortization

Initialize $\phi$
**for** each environment step **do**
    $\boldsymbol{\lambda} \leftarrow f_\phi(\mathbf{s}_t)$
    $\mathbf{a}_t \sim \pi_\phi(\mathbf{a}_t|\mathbf{s}_t, \mathcal{O}; \boldsymbol{\lambda})$
    $\mathbf{s}_{t+1} \sim p_{\text{env}}(\mathbf{s}_{t+1}|\mathbf{s}_t, \mathbf{a}_t)$
**end for**
**for** each training step **do**
    $\phi \leftarrow \phi + \eta \nabla_\phi \mathcal{J}$
**end for**

**Algorithm 2** Iterative Amortization

Initialize $\phi$
**for** each environment step **do**
    Initialize $\boldsymbol{\lambda}$
    **for** each policy optimization iteration **do**
        $\boldsymbol{\lambda} \leftarrow f_\phi(\mathbf{s}_t, \boldsymbol{\lambda}, \nabla_{\boldsymbol{\lambda}} \mathcal{J})$
    **end for**
    $\mathbf{a}_t \sim \pi_\phi(\mathbf{a}_t|\mathbf{s}_t, \mathcal{O}; \boldsymbol{\lambda})$
    $\mathbf{s}_{t+1} \sim p_{\text{env}}(\mathbf{s}_{t+1}|\mathbf{s}_t, \mathbf{a}_t)$
**end for**
**for** each training step **do**
    $\phi \leftarrow \phi + \eta \nabla_\phi \mathcal{J}$
**end for**

We see that the estimate of a direct amortized policy (diamond) is suboptimal, far from the optimal estimate (star). Additional 2D plots are shown in Figure B.3. However, note that the absolute difference in the objective due to direct amortization is relatively small compared with the objective itself. That is, suboptimal estimates tend to have only a *minor* impact on evaluation performance, as we show in Appendix B.4. Rather, policy suboptimality hinders data collection, sampling fewer actions with high value estimates. Further, direct amortization is typically limited to a *single* static estimate of the policy, unable to directly adapt to the RL objective and therefore limiting exploration. To improve upon this scheme, in Section 3, we turn to a technique developed in generative modeling, *iterative amortization* (Marino et al., 2018b), which retains the efficiency benefits of amortization while employing a more flexible iterative estimation procedure.

## 2.4 RELATED WORK

Previous works have investigated methods for improving policy optimization. QT-Opt (Kalash-nikov et al., 2018) uses the cross-entropy method (CEM) (Rubinstein & Kroese, 2013), an iterative derivative-free optimizer, to optimize a $Q$-value estimator for robotic grasping. CEM and related methods are also used in model-based RL for performing model-predictive control (Nagabandi et al., 2018; Chua et al., 2018; Piché et al., 2019; Hafner et al., 2019). Gradient-based policy optimization, in contrast, is less common (Henaff et al., 2017; Srinivas et al., 2018; Bharadhwaj et al., 2020), however, gradient-based optimization can also be combined with CEM (Amos & Yarats, 2020). Most policy-based methods use direct amortization, either using a feedforward (Haarnoja et al., 2018b) or recurrent (Guez et al., 2019) network. Similar approaches have also been applied to model-based value estimates (Byravan et al., 2020; Clavera et al., 2020; Amos et al., 2020), as well as combining direct amortization with model predictive control (Lee et al., 2019) and planning (Rivière et al., 2020). A separate line of work has explored improving the policy distribution, using normalizing flows (Haarnoja et al., 2018a; Tang & Agrawal, 2018) and latent variables (Tirumala et al., 2019). In principle, iterative amortization can perform policy optimization in each of these settings.

## 3 ITERATIVE AMORTIZED POLICY OPTIMIZATION

### 3.1 FORMULATION

Iterative amortized optimizers (Marino et al., 2018b) utilize some form of error or gradient to update the approximate posterior distribution parameters. While various forms exist, we consider gradient-encoding models (Andrychowicz et al., 2016) due to their generality. Compared with direct amortization in Eq. 9, we use iterative amortized optimizers of the general form

$$\boldsymbol{\lambda} \leftarrow f_\phi(\mathbf{s}, \boldsymbol{\lambda}, \nabla_{\boldsymbol{\lambda}} \mathcal{J}), \qquad \text{(iterative amortization)} \qquad (10)$$

also shown in Figure 1 (Right), where $f_\phi$ is a deep network and $\boldsymbol{\lambda}$ are the action distribution parameters. For example, if $\pi = \mathcal{N}(\mathbf{a}; \boldsymbol{\mu}, \text{diag}(\boldsymbol{\sigma}^2))$, then $\boldsymbol{\lambda} \equiv [\boldsymbol{\mu}, \boldsymbol{\sigma}]$. Technically, $\mathbf{s}$ is redundant, as the state dependence is already captured in $\mathcal{J}$, but this can empirically improve performance (Marino

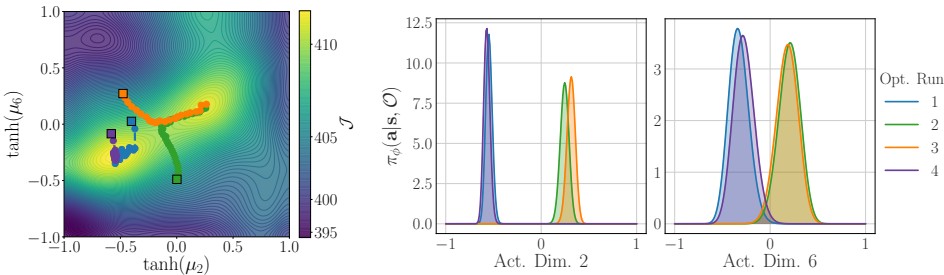

Figure 2: **Estimating Multiple Policy Modes**. Unlike direct amortization, which is restricted to a single estimate, iterative amortization can effectively sample from multiple high-value action modes. This is shown for a particular state in `Ant-v2`, showing multiple optimization runs across two action dimensions (**Left**). Each square denotes an initialization. The optimizer finds both modes, with the densities plotted on the **Right**. This capability provides increased flexibility in action exploration.

et al., 2018b). In practice, the update is carried out using a "highway" gating operation (Hochreiter & Schmidhuber, 1997; Srivastava et al., 2015). Denoting $\omega_\phi \in [0, 1]$ as the gate and $\delta_\phi$ as the update, both of which are output by $f_\phi$, the gating operation is expressed as

$$\lambda \leftarrow \omega_\phi \odot \lambda + (1 - \omega_\phi) \odot \delta_\phi, \tag{11}$$

where $\odot$ denotes element-wise multiplication. This update is typically run for a fixed number of steps, and, as with a direct policy, the iterative optimizer is trained using stochastic gradient estimates of $\nabla_\phi \mathcal{J}$, obtained through the path-wise derivative estimator (Kingma & Welling, 2014; Rezende et al., 2014; Heess et al., 2015). Because the gradients $\nabla_\lambda \mathcal{J}$ must be estimated online, i.e. during policy optimization, this scheme requires some way of estimating $\mathcal{J}$ online, e.g. through a parameterized $Q$-value network (Mnih et al., 2013) or a differentiable model (Heess et al., 2015).

## 3.2 CONSIDERATIONS

### 3.2.1 ADDED FLEXIBILITY

Iterative amortized optimizers are more flexible than their direct counterparts, incorporating feedback from the objective *during* policy optimization (Algorithm 2), rather than only *after* optimization (Algorithm 1). Increased flexibility improves the accuracy of optimization, thereby tightening the variational bound (Marino et al., 2018b;a). We see this flexibility in Figure 1 (Left), where an iterative amortized policy network, despite being trained with a *different* value estimator, is capable of iteratively optimizing the policy estimate (blue dots), quickly arriving near the optimal estimate.

Direct amortization is typically restricted to a single estimate, inherently limiting exploration. In contrast, iterative amortized optimizers, by using stochastic gradients and random initialization, can traverse the optimization landscape. As with any iterative optimization scheme, this allows iterative amortization to obtain multiple valid estimates (Greff et al., 2019). We illustrate this capability across two action dimensions in Figure 2 for a state in the `Ant-v2` MuJoCo environment. Over multiple policy optimization runs, iterative amortization finds multiple modes, sampling from two high-value regions of the action space. This provides increased flexibility in action exploration.

### 3.2.2 MITIGATING VALUE OVERESTIMATION

Model-free approaches generally estimate $Q_\pi$ using function approximation and temporal difference learning. However, this comes with the pitfall of value overestimation, i.e. positive bias in the estimate, $\widehat{Q}_\pi$ (Thrun & Schwartz, 1993). This issue is tied to uncertainty in the value estimate, though it is distinct from optimism under uncertainty. If the policy can exploit regions of high uncertainty, the resulting target values will introduce positive bias into the estimate. More flexible policy optimizers may exacerbate the problem, exploiting this uncertainty to a greater degree. Further, a rapidly changing policy increases the difficulty of value estimation (Rajeswaran et al., 2020).

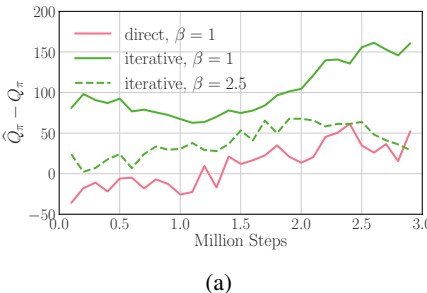 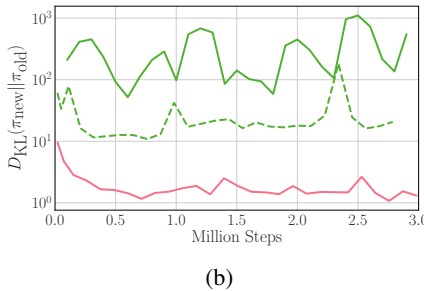

(a)                                              (b)

Figure 3: **Mitigating Value Overestimation**. Using the same value estimation setup ($\beta = 1$ in Eq. 12), shown on `Ant-v2`, iterative amortization results in (**a**) higher value overestimation bias (closer to zero is better) and (**b**) a more rapidly changing policy as compared with direct amortization. Increasing $\beta$ helps to mitigate these issues by further penalizing variance in the value estimate.

Various techniques have been proposed for mitigating value overestimation in deep RL. The most prominent technique, double deep $Q$-network (Van Hasselt et al., 2016) maintains two $Q$-value estimates (Van Hasselt, 2010), attempting to decouple policy optimization from value estimation. Fujimoto et al. (2018) apply and improve upon this technique for actor-critic settings, estimating the target $Q$-value as the minimum of two $Q$-networks, $Q_{\psi_1}$ and $Q_{\psi_2}$:

$$\widehat{Q}_\pi(\mathbf{s}, \mathbf{a}) = \min_{i=1,2} Q_{\psi_i'}(\mathbf{s}, \mathbf{a}),$$

where $\psi_i'$ denotes the "target" network parameters. As noted by Fujimoto et al. (2018), this not only counteracts value overestimation, but also penalizes high-variance value estimates, because the minimum decreases with the variance of the estimate. Ciosek et al. (2019) noted that, for a bootstrapped ensemble of two $Q$-networks, the minimum operation can be interpreted as estimating

$$\widehat{Q}_\pi(\mathbf{s}, \mathbf{a}) = \mu_Q(\mathbf{s}, \mathbf{a}) - \beta\sigma_Q(\mathbf{s}, \mathbf{a}), \tag{12}$$

with mean $\mu_Q(\mathbf{s}, \mathbf{a}) \equiv \frac{1}{2}\sum_{i=1,2} Q_{\psi_i'}(\mathbf{s}, \mathbf{a})$, standard deviation $\sigma_Q(\mathbf{s}, \mathbf{a}) \equiv (\frac{1}{2}\sum_{i=1,2}(Q_{\psi_i'}(\mathbf{s}, \mathbf{a}) - \mu_Q(\mathbf{s}, \mathbf{a}))^2)^{1/2}$, and $\beta = 1$. Thus, to further penalize high-variance value estimates, preventing value overestimation, we can increase $\beta$. For large $\beta$, however, value estimates become overly pessimistic, negatively impacting training. Thus, $\beta$ reduces target value variance at the cost of increased bias.

Due to the flexibility of iterative amortization, the default $\beta = 1$ results in increased value bias (Figure 3a) and a more rapidly changing policy (Figure 3b) as compared with direct amortization. Further penalizing high-variance target values with $\beta = 2.5$ reduces value overestimation and improves policy stability. For details, see Appendix A.2. Recent techniques for mitigating overestimation have been proposed, such as adjusting the temperature, $\alpha$ (Fox, 2019). In offline RL, this issue has been tackled through the action prior (Fujimoto et al., 2019; Kumar et al., 2019; Wu et al., 2019) or by altering $Q$-network training (Agarwal et al., 2019; Kumar et al., 2020). While such techniques could be used here, increasing $\beta$ provides a simple solution with no additional computational overhead.

## 4 EXPERIMENTS

### 4.1 SETUP

To focus on policy optimization, we implement iterative amortized policy optimization using the soft actor-critic (SAC) setup described by Haarnoja et al. (2018c). This uses two $Q$-networks, uniform action prior, $p_\theta(\mathbf{a}|\mathbf{s}) = \mathcal{U}(-1, 1)$, and a tuning scheme for the temperature, $\alpha$. In our experiments, "direct" refers to direct amortization employed in SAC, i.e. a direct policy network, and "iterative" refers to iterative amortization. Both approaches use the *same* network architecture, adjusting only the number of inputs and outputs to accommodate gradients, current policy estimates, and gated updates (Sec. 3.1). Unless otherwise stated, we use 5 iterations per time step for iterative amortization, following Marino et al. (2018b). For details, refer to Appendix A and Haarnoja et al. (2018b;c).

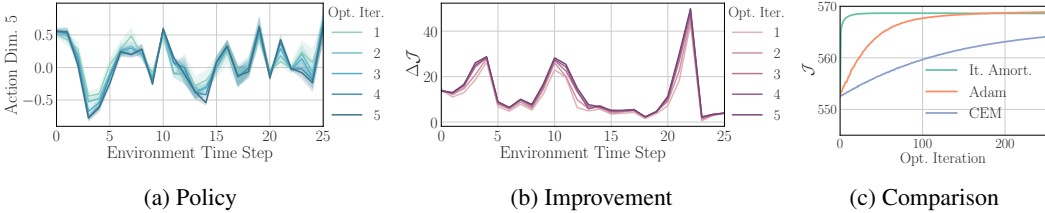

(a) Policy        (b) Improvement        (c) Comparison

Figure 4: **Policy Optimization**. Visualization over time steps of **(a)** one dimension of the policy distribution and **(b)** the improvement in the objective, $\Delta \mathcal{J}$, across policy optimization iterations. **(c)** Comparison of iterative amortization with Adam (Kingma & Ba, 2014) (gradient-based) and CEM (Rubinstein & Kroese, 2013) (gradient-free). Iterative amortization is substantially more efficient.

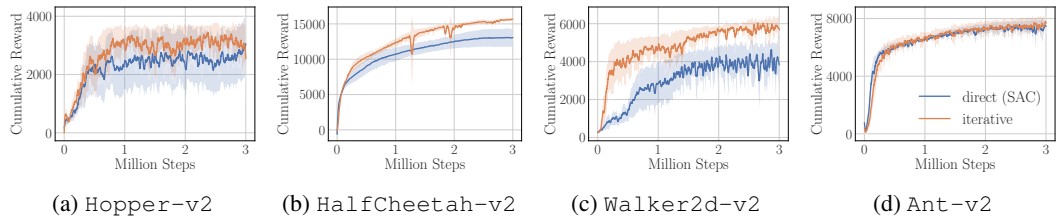

(a) `Hopper-v2`    (b) `HalfCheetah-v2`    (c) `Walker2d-v2`    (d) `Ant-v2`

Figure 5: **Performance Comparison**. Iterative amortized policy optimization performs comparably with or better than direct amortized policies across a range of MuJoCo environments. Performance curves show the mean and $\pm$ standard deviation over 5 random seeds.

## 4.2 ANALYSIS

### 4.2.1 VISUALIZING POLICY OPTIMIZATION

In Figure 1 (Left), we visualize the trajectory of iterative amortized policy optimization along two dimensions of the policy mean on a state from `Hopper-v2`. Through iterative optimization, the network arrives near the optimum. Notably, this is performed with a value function trained using a *different*, direct policy, demonstrating generalization to other optimization landscapes. Additional 2D plots comparing direct and iterative amortization are shown in Figure B.3 in the appendix. In Figure 4, we visualize iterative refinement using a single action dimension from `Ant-v2` across time steps. The refinements in Figure 4a give rise to the objective improvements in Figure 4b. We compare with Adam (Kingma & Ba, 2014) (gradient-based) and CEM (Rubinstein & Kroese, 2013) (gradient-free) in Figure 4c (see Appendix B.2), where iterative amortization is substantially more efficient. This trend is consistent across environments, as shown in Figure B.2 in the appendix.

### 4.2.2 PERFORMANCE COMPARISON

We evaluate iterative amortized policy optimization on MuJoCo (Todorov et al., 2012) continuous control tasks from OpenAI gym (Brockman et al., 2016). In Figure 5, we compare the cumulative reward of direct and iterative amortized policy optimization across environments. Each curve shows the mean and $\pm$ standard deviation of 5 random seeds. In all cases, iterative amortized policy optimization matches or outperforms the baseline direct method, both in sample efficiency and final performance. Across environments, iterative amortization also yields more consistent performance.

### 4.2.3 DECREASED AMORTIZATION GAP

To evaluate policy optimization, we estimate per-step amortization gaps using the experiments from Figure 5, performing additional iterations of gradient ascent on $\mathcal{J}$, w.r.t. the policy parameters, $\boldsymbol{\lambda} \equiv [\boldsymbol{\mu}, \boldsymbol{\sigma}]$ (see Appendix A.3). We also evaluate the iterative agents trained with 5 iterations for an additional 5 amortized iterations. Results are shown in Figure 6. We emphasize that it is challenging to *directly* compare amortization gaps across optimization schemes, as these involve different value functions, and therefore different objectives. Likewise, we estimate the amortization gap using the learned $Q$-value networks, which may be biased (Figure 3). Nevertheless, we find that iterative

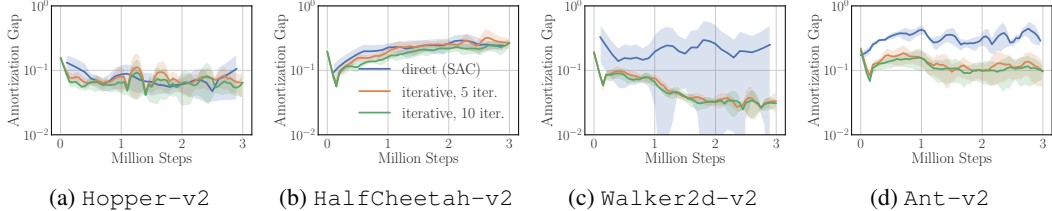

Figure 6: **Decreased Amortization Gap**. Estimated amortization gaps per step for direct and iterative amortized policy optimization. Iterative amortization achieves comparable or lower gaps across environments. Gaps are estimated using stochastic gradient-based optimization over 100 random states. Curves show the mean and $\pm$ standard deviation over 5 random seeds.

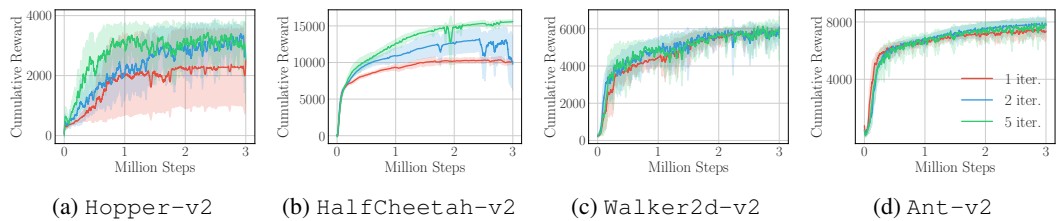

Figure 7: **Iterations During Training**. Performance of iterative amortized policy optimization for varying numbers of iterations during training. Increasing the number of iterations generally results in improvements. Curves show the mean and $\pm$ standard deviation over 4 random seeds.

amortized policy optimization achieves, on average, lower amortization gaps than direct amortization across all environments. Further amortized iterations at evaluation yield further estimated improvement, demonstrating generalization.

However, we note that the amortization gaps are relatively small compared to the estimated discounted objective. Accordingly, when we evaluate the more fully optimized policies in the environment, we do not observe a noticeable increase in performance (see Appendix B.4). This demonstrates that policy suboptimality is not a significant concern for *evaluation*. Rather, improved policy optimization is helpful for *training*. This allows the agent to collect data where value estimates are highest and ultimately improve these value estimates. Indeed, when we train iterative amortization while varying the policy optimization iterations per step (Section 4.2.4), we observe that the estimated amortization gap again decreases with increasing iterations, but now with a corresponding increase in performance (Figure B.6). Thus, reducing the amortization gap only *indirectly* improves task performance by improving training, with the relationship depending on the $Q$-value estimator and other factors.

### 4.2.4 VARYING ITERATIONS

Direct amortized policy optimization is restricted to a single forward pass through the network. Iterative amortization, in contrast, is capable of improving during policy optimization with additional computation time. To demonstrate this capability, we train iterative amortized policy optimization while varying the number of iterations in $\{1, 2, 5\}$. In Figure 7, we see that increasing the number of amortized optimization iterations generally improves sample efficiency (`Walker2d-v2`), asymptotic performance (`Ant-v2`), or both (`Hopper-v2` & `HalfCheetah-v2`). Thus, the quality of the policy optimizer can play a significant role in determining performance through training.

### 4.2.5 ITERATIVE AMORTIZATION WITH MODEL-BASED VALUE ESTIMATES

While our analysis has centered on the model-free setting, iterative amortized policy optimization can also be applied to model-based value estimates. As model-based RL remains an active research area (Janner et al., 2019), we provide a proof-of-concept in this setting, using a learned deterministic model on `HalfCheetah-v2` (see Appendix A.5). As shown in Figure 8a, iterative amortization outperforms direct amortization in this setting. Iterative amortization refines planned trajectories,

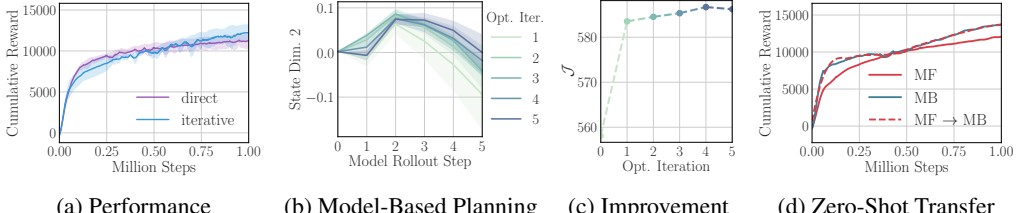

(a) Performance     (b) Model-Based Planning     (c) Improvement     (d) Zero-Shot Transfer

Figure 8: **Optimizing Model-Based Value Estimates**. **(a)** Performance comparison of direct and iterative amortization using model-based value estimates. **(b)** Planned trajectories over policy optimization iterations. **(c)** The corresponding estimated objective increases over iterations. **(d)** Zero-shot transfer of iterative amortization from model-free (MF) to model-based (MB) estimates.

shown for a single state dimension in Figure 8b, yielding corresponding improvements (Figure 8c). Further, because we are learning an iterative policy optimizer, we can zero-shot transfer a policy optimizer trained with a model-free value estimator to a model-based value estimator (Figure 8d). This is not possible with a direct amortized optimizer, which does not use value estimates online during policy optimization. Iterative amortization is capable of generalizing to new value estimates, *instantly* incorporating updated value estimates in policy optimization. This demonstrates and highlights the opportunity for improving model-based planning through iterative amortization.

## 5    DISCUSSION

We have introduced iterative amortized policy optimization, a flexible and powerful policy optimization technique. Using the MuJoCo continuous control suite, we have demonstrated improved performance over direct amortization with both model-based and model-free value estimates. Iterative amortization provides a drop-in replacement and improvement over direct policy networks in deep RL. Although iterative amortized policy optimization requires additional computation, this could be combined with some form of adaptive computation time (Graves, 2016; Figurnov et al., 2018), gauging the required iterations. Likewise, efficiency depends, in part, on the policy initialization, which could be improved by learning the action prior, $p_\theta(\mathbf{a}|\mathbf{s})$ (Abdolmaleki et al., 2018). The power of iterative amortization is in using the value estimate during policy optimization to iteratively improve the policy *online*. This is a form of negative feedback control (Astrom & Murray, 2008), using errors to guide policy optimization. Beyond providing a more powerful optimizer, we are hopeful that iterative amortized policy optimization, by using online feedback, will enable a range of improved RL algorithms, capable of instantly adapting to different value estimates, as shown in Figure 8d.

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

## A  EXPERIMENT DETAILS

### A.1  2D PLOTS

In Figures 1 and 2, we plot the estimated variational objective, $\mathcal{J}$, as a function of two dimensions of the policy mean, $\boldsymbol{\mu}$. To create these plots, we first perform policy optimization (direct amortization in Figure 1 and iterative amortization in Figure 2), estimating the policy mean and variance. This is performed using on-policy trajectories from evaluation episodes (for a direct agent in Figure 1 and an iterative agent in Figure 2). While holding all other dimensions of the policy constant, we then estimate the variational objective while varying two dimensions of the mean (1 & 3 in Figure 1 and 2 & 6 in Figure 2). Iterative amortization is additionally performed while preventing any updates to the constant dimensions. Even in this restricted setting, iterative amortization is capable of optimizing the policy. Additional 2D plots comparing direct vs. iterative amortization on other environments are shown in Figure 14, where we see similar trends.

### A.2  VALUE BIAS ESTIMATION

We estimate the bias in the $Q$-value estimator using a similar procedure as Fujimoto et al. (2018), comparing the estimate of the $Q$-networks ($\widehat{Q}_\pi$) with a Monte Carlo estimate of the future objective in the actual environment, $Q_\pi$, using a set of state-action pairs. To enable comparison across setups, we collect 100 state-action pairs using a uniform random policy, then evaluate the estimator's bias, $\mathbb{E}_{\mathbf{s},\mathbf{a}}\left[\widehat{Q}_\pi - Q_\pi\right]$, throughout training. To obtain the Monte Carlo estimate of $Q_\pi$, we use 100 action samples, which are propagated through all future time steps. The result is discounted using the same discounting factor as used during training, $\gamma = 0.99$, as well as the same Lagrange multiplier, $\alpha$. Figure 3 shows the mean and $\pm$ standard deviation across the 100 state-action pairs.

### A.3  AMORTIZATION GAP ESTIMATION

Calculating the amortization gap in the RL setting is challenging, as properly evaluating the variational objective, $\mathcal{J}$, involves unrolling the environment. During training, the objective is estimated using a set of $Q$-networks and/or a learned model. However, finding the optimal policy distribution, $\widehat{\pi}$, under these learned value estimates may not accurately reflect the amortization gap, as the value estimator likely contains positive bias (Figure 3). Because the value estimator is typically locally accurate near the current policy, we estimate the amortization gap by performing gradient ascent on $\mathcal{J}$ w.r.t. the policy distribution parameters, $\boldsymbol{\lambda}$, initializing from the amortized estimate (from $\pi_\phi$). This is a form *semi-amortized* variational inference (Hjelm et al., 2016; Krishnan et al., 2018; Kim et al., 2018). We use the Adam optimizer (Kingma & Ba, 2014) with a learning rate of $5 \times 10^{-3}$ for 100 gradient steps, which we found consistently converged. This results in the estimated optimized $\widehat{\pi}$. We estimate the gap using 100 on-policy states, calculating $\mathcal{J}(\theta, \widehat{\pi}) - \mathcal{J}(\theta, \pi)$, i.e. the improvement in the objective after gradient-based optimization. Figure 6 shows the resulting mean and $\pm$ standard deviation. We also run iterative amortized policy optimization for an additional 5 iterations during this evaluation, empirically yielding an additional decrease in the estimated amortization gap.

### A.4  HYPERPARAMETERS

Our setup follows that of soft actor-critic (SAC) (Haarnoja et al., 2018b;c), using a uniform action prior, i.e. entropy regularization, and two $Q$-networks (Fujimoto et al., 2018). Off-policy training is performed using a replay buffer (Lin, 1992; Mnih et al., 2013). Training hyperparameters are given in Table 6.

**Temperature**  Following Haarnoja et al. (2018c), we adjust the temperature, $\alpha$, to maintain a specified entropy constraint, $\epsilon_\alpha = |\mathcal{A}|$, where $|\mathcal{A}|$ is the size of the action space, i.e. the dimensionality.

**Policy**  We use the same network architecture (number of layers, units/layer, non-linearity) for both direct and iterative amortized policy optimizers (Table 2). Each policy network results in Gaussian distribution parameters, and we apply a `tanh` transform to ensure $\mathbf{a} \in [-1, 1]$ (Haarnoja et al., 2018b). In the case of a Gaussian, the distribution parameters are $\boldsymbol{\lambda} = [\boldsymbol{\mu}, \boldsymbol{\sigma}]$. The inputs and

Table 1: **Policy Inputs & Outputs**.

|          | Inputs | Outputs |
| -------- | :----: | :-----: |
| Direct   | $\mathbf{s}$ | $\boldsymbol{\lambda}$ |
| Iterative | $\mathbf{s}, \boldsymbol{\lambda}, \nabla_{\boldsymbol{\lambda}}\mathcal{J}$ | $\boldsymbol{\delta}, \boldsymbol{\omega}$ |

Table 2: **Policy Networks**.

| Hyperparameter | Value |
| -------------- | ----: |
| Number of Layers | 2 |
| Number of Units / Layer | 256 |
| Non-linearity | ReLU |

Direct

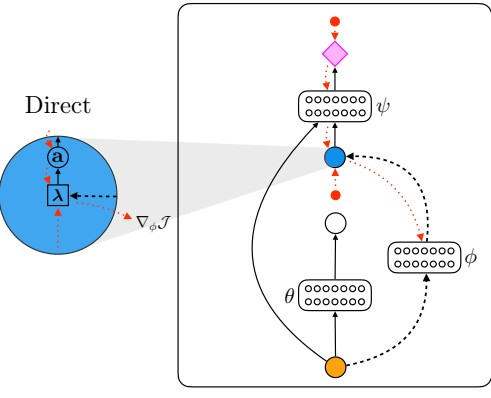

Iterative

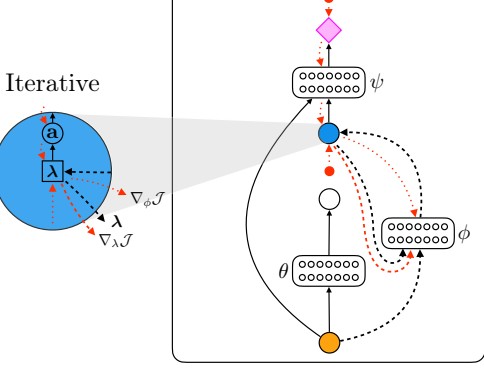

(a) Direct Amortization        (b) Iterative Amortization

Figure 9: **Amortized Optimizers**. Diagrams of **(a)** direct and **(b)** iterative amortized policy optimization. As in Figure 1, larger circles represent probability distributions, and smaller red circles represent terms in the objective. Red dotted arrows represent gradients. In addition to the state, $\mathbf{s}_t$, iterative amortization uses the current policy distribution estimate, $\boldsymbol{\lambda}$, and the policy optimization gradient, $\nabla_{\boldsymbol{\lambda}}\mathcal{J}$, to iteratively optimize $\mathcal{J}$. Like direct amortization, the optimizer network parameters, $\phi$, are updated using $\nabla_{\phi}\mathcal{J}$. This generally requires some form of stochastic gradient estimation to differentiate through $\mathbf{a}_t \sim \pi(\mathbf{a}_t|\mathbf{s}_t, \mathcal{O}; \boldsymbol{\lambda})$.

outputs of each optimizer form are given in Table 1. Again, $\boldsymbol{\delta}$ and $\boldsymbol{\omega}$ are respectively the update and gate of the iterative amortized optimizer (Eq. 11), each of which are defined for both $\boldsymbol{\mu}$ and $\boldsymbol{\sigma}$. Following Marino et al. (2018b), we apply layer normalization (Ba et al., 2016) individually to each of the inputs to iterative amortized optimizers. We initialize iterative amortization with $\boldsymbol{\mu} = \mathbf{0}$ and $\boldsymbol{\sigma} = \mathbf{1}$, however, these could be initialized from a learned action prior (Marino et al., 2018a).

Table 3: $Q$-**value Network Architecture A**.

| Hyperparameter | Value |
| -------------- | ----: |
| Number of Layers | 2 |
| Number of Units / Layer | 256 |
| Non-linearity | ReLU |
| Layer Normalization | False |
| Connectivity | Sequential |

Table 4: $Q$-**value Network Architecture B**.

| Hyperparameter | Value |
| -------------- | ----: |
| Number of Layers | 3 |
| Number of Units / Layer | 512 |
| Non-linearity | ELU |
| Layer Normalization | True |
| Connectivity | Highway |

$Q$-**value** We investigated two $Q$-value network architectures. Architecture A (Table 3) is the same as that from Haarnoja et al. (2018b). Architecture B (Table 4) is a wider, deeper network with highway connectivity (Srivastava et al., 2015), layer normalization (Ba et al., 2016), and ELU non-linearities (Clevert et al., 2015). We initially compared each $Q$-value network architecture using each policy optimizer on each environment, as shown in Figure 10. The results in Figure 5 were obtained using the better performing architecture in each case, given in Table 5. As in Fujimoto et al. (2018), we use an ensemble of 2 separate $Q$-networks in each experiment.

**Value Pessimism ($\beta$)** As discussed in Section 3.2.2, the increased flexibility of iterative amortization allows it to potentially exploit inaccurate value estimates. We increased the pessimism hyperpa-

Table 5: $Q$-**value Network Architecture by Environment**.

|          | Hopper-v2 | HalfCheetah-v2 | Walker2d-v2 | Ant-v2 |
|----------|-----------|----------------|-------------|--------|
| Direct   | A         | B              | A           | B      |
| Iterative| A         | A              | B           | B      |

(a) Hopper-v2     (b) HalfCheetah-v2     (c) Walker2d-v2     (d) Ant-v2

Figure 10: **Value Architecture Comparison**. Plots show performance for $\geq 3$ seeds for each value architecture (A or B) for each policy optimization technique (direct or iterative). Note: results for iterative + B on Hopper-v2 were obtained with an overly pessimistic value estimate ($\beta = 2.5$ rather than $\beta = 1.5$) and are consequently worse.

rameter, $\beta$, to further penalize variance in the value estimate. Experiments with direct amortization use the default $\beta = 1$ in all environments, as we did not find that increasing $\beta$ helped in this setup. For iterative amortization, we use $\beta = 1.5$ on Hopper-v2 and $\beta = 2.5$ on all other environments. This is only applied during training; while collecting data in the environment, we use $\beta = 1$ to not overly penalize exploration.

Table 6: **Training Hyperparameters**.

| Hyperparameter | Value |
|----------------|-------|
| Discount Factor $(\gamma)$ | 0.99 |
| $Q$-network Update Rate $(\tau)$ | $5 \cdot 10^{-3}$ |
| Network Optimizer | Adam |
| Learning Rate | $3 \cdot 10^{-4}$ |
| Batch Size | 256 |
| Initial Random Steps | $5 \cdot 10^3$ |
| Replay Buffer Size | $10^6$ |

## A.5 MODEL-BASED VALUE ESTIMATION

For model-based experiments, we use a single, deterministic model together with the ensemble of 2 $Q$-value networks (discussed above).

**Model** We use separate networks to estimate the state transition dynamics, $p_{\text{env}}(\mathbf{s}_{t+1}|\mathbf{s}_t, \mathbf{a}_t)$, and reward function, $r(\mathbf{s}_t, \mathbf{a}_t)$. The network architecture is given in Table 7. Each network outputs the mean of a Gaussian distribution; the standard deviation is a separate, learnable parameter. The reward network directly outputs the mean estimate, whereas the state transition network outputs a residual estimate, $\Delta_{\mathbf{s}_t}$, yielding an updated mean estimate through:

$$\boldsymbol{\mu}_{\mathbf{s}_{t+1}} = \mathbf{s}_t + \Delta_{\mathbf{s}_t}.$$

**Model Training** The state transition and reward networks are both trained using maximum log-likelihood training, using data examples from the replay buffer. Training is performed at the same frequency as policy and $Q$-network training, using the same batch size (256) and network optimizer. However, we perform $10^3$ updates at the beginning of training, using the initial random steps, in order to start with a reasonable model estimate.

Table 7: **Model Network Architectures**.

| Hyperparameter | Value |
|---|---|
| Number of Layers | 2 |
| Number of Units / Layer | 256 |
| Non-linearity | `Leaky ReLU` |
| Layer Normalization | True |

Table 8: **Model-Based Hyperparameters**.

| Hyperparameter | Value |
|---|---|
| Rollout Horizon, $h$ | 2 |
| Retrace $\lambda$ | 0.9 |
| Pre-train Model Updates | $10^3$ |
| Model-Based Value Targets | True |

**Value Estimation**  To estimate $Q$-values, we combine short model rollouts with the model-free estimates from the $Q$-networks. Specifically, we unroll the model and policy, obtaining state, reward, and policy estimates at current and future time steps. We then apply the $Q$-value networks to these future state-action estimates. Future rewards and value estimates are combined using the Retrace estimator (Munos et al., 2016). Denoting the estimate from the $Q$-network as $\widehat{Q}_\psi(\mathbf{s}, \mathbf{a})$ and the reward estimate as $\widehat{r}(\mathbf{s}, \mathbf{a})$, we calculate the $Q$-value estimate at the current time step as

$$\widehat{Q}_\pi(\mathbf{s}_t, \mathbf{a}_t) = \widehat{Q}_\psi(\mathbf{s}_t, \mathbf{a}_t) + \mathbb{E}\left[\sum_{t'=t}^{t+h} \gamma^{t'-t}\lambda^{t'-t}\left(\widehat{r}(\mathbf{s}_{t'}, \mathbf{a}_{t'}) + \gamma\widehat{V}_\psi(\mathbf{s}_{t'+1}) - \widehat{Q}_\psi(\mathbf{s}_{t'}, \mathbf{a}_{t'})\right)\right], \quad (13)$$

where $\lambda$ is an exponential weighting factor, $h$ is the rollout horizon, and the expectation is evaluated under the model and policy. In the variational RL setting, the state-value, $V_\pi(\mathbf{s})$, is

$$V_\pi(\mathbf{s}) = \mathbb{E}_\pi\left[Q_\pi(\mathbf{s}, \mathbf{a}) - \alpha \log \frac{\pi(\mathbf{a}|\mathbf{s}, \mathcal{O})}{p_\theta(\mathbf{a}|\mathbf{s})}\right]. \quad (14)$$

In Eq. 13, we approximate $V_\pi$ using the $Q$-network to approximate $Q_\pi$ in Eq. 14, yielding $\widehat{V}_\psi(\mathbf{s})$. Finally, to ensure consistency between the model and the $Q$-value networks, we use the model-based estimate from Eq. 13 to provide target values for the $Q$-networks, as in Janner et al. (2019).

**Future Policy Estimates**  Evaluating the expectation in Eq. 13 requires estimates of $\pi$ at future time steps. This is straightforward with direct amortization, which employs a feedforward policy, however, with iterative amortization, this entails recursively applying an iterative optimization procedure. Alternatively, we could use the prior, $p_\theta(\mathbf{a}|\mathbf{s})$, at future time steps, but this does not apply in the max-entropy setting, where the prior is uniform. For computational efficiency, we instead learn a separate direct (amortized) policy for model-based rollouts. That is, with iterative amortization, we create a separate direct network using the same hyperparameters from Table 2. This network distills iterative amortization into a direct amortized optimizer, through the KL divergence, $D_{\text{KL}}(\pi_{\text{it.}}||\pi_{\text{dir.}})$. Rollout policy networks are common in model-based RL (Silver et al., 2016; Piché et al., 2019).

## B  ADDITIONAL RESULTS

### B.1  IMPROVEMENT PER STEP

In Figure 12, we plot the average improvement in the variational objective per step throughout training, with each curve showing a different random seed. That is, each plot shows the average change in the variational objective after running 5 iterations of iterative amortized policy optimization. With the exception of `HalfCheetah-v2`, the improvement remains relatively constant throughout training and consistent across seeds.

### B.2  COMPARISON WITH ITERATIVE OPTIMIZERS

Iterative amortized policy optimization obtains the *accuracy* benefits of iterative optimization while retaining the *efficiency* benefits of amortization. In Section 4, we compared the accuracy of iterative and direct amortization, seeing that iterative amortization yields reduced amortization gaps (Figure 6) and improved performance (Figure 5). In this section, we compare iterative amortization with two popular iterative optimizers: Adam (Kingma & Ba, 2014), a gradient-based optimizer, and cross-entropy method (CEM) (Rubinstein & Kroese, 2013), a gradient-free optimizer.

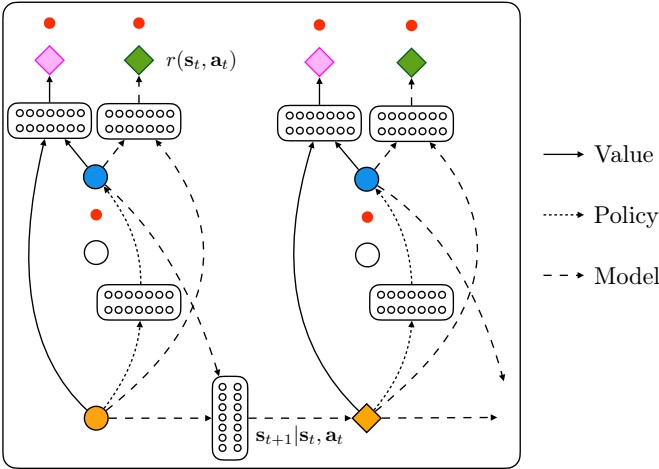

Figure 11: **Model-Based Value Estimation**. Diagram of model-based value estimation (shown with direct amortization). For clarity, the diagram is shown without the policy prior network, $p_\theta(\mathbf{a}_t|\mathbf{s}_t)$. The model consists of a deterministic reward estimate, $r(\mathbf{s}_t, \mathbf{a}_t)$, (green diamond) and a state estimate, $\mathbf{s}_{t+1}|\mathbf{s}_t, \mathbf{a}_t$, (orange diamond). The model is unrolled over a horizon, $H$, and the $Q$-value is estimated using the Retrace estimator (Munos et al., 2016), given in Eq. 13.

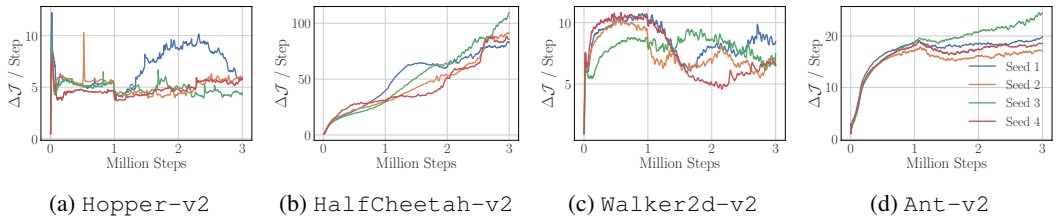

(a) Hopper-v2    (b) HalfCheetah-v2    (c) Walker2d-v2    (d) Ant-v2

Figure 12: **Per-Step Improvement**. Each plot shows the per-step improvement in the estimated variational RL objective, $\mathcal{J}$, throughout training resulting from iterative amortized policy optimization. Each curve denotes a different random seed.

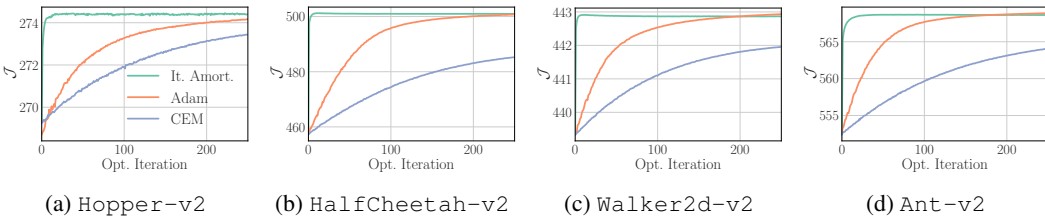

(a) Hopper-v2    (b) HalfCheetah-v2    (c) Walker2d-v2    (d) Ant-v2

Figure 13: **Comparison with Iterative Optimizers**. Average estimated objective over policy optimization iterations, comparing with Adam (Kingma & Ba, 2014) and CEM (Rubinstein & Kroese, 2013). These iterative optimizers require over an order of magnitude more iterations to reach comparable performance with iterative amortization, making them impractical in many applications.

To compare the accuracy and efficiency of the optimizers, we collect 100 states for each seed in each environment from the model-free experiments in Section 4.2.2. For each optimizer, we optimize the variational objective, $\mathcal{J}$, starting from the same initialization. Tuning the step size, we found that 0.01 yielded the steepest improvement without diverging for both Adam and CEM. Gradients are evaluated with 10 action samples. For CEM, we sample 100 actions and fit a Gaussian mean and variance to the top 10 samples. This is comparable with QT-Opt (Kalashnikov et al., 2018), which draws 64 samples and retains the top 6 samples.

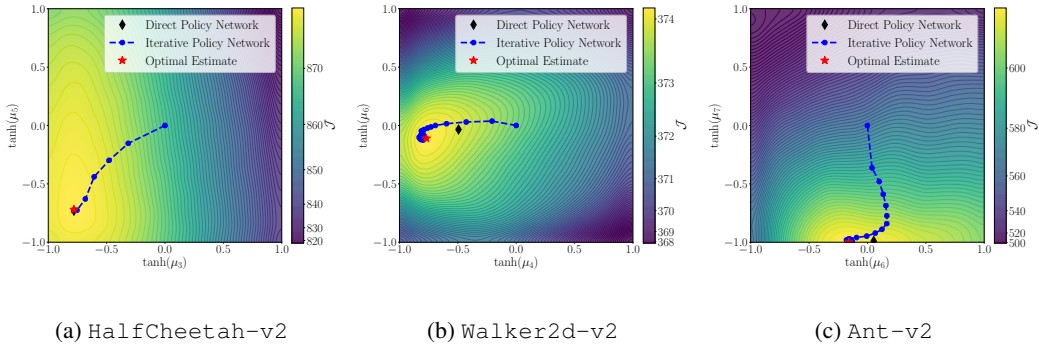

(a) `HalfCheetah-v2`          (b) `Walker2d-v2`          (c) `Ant-v2`

Figure 14: **2D Optimization Plots**. Each plot shows the optimization objective over two dimensions of the policy mean, $\mu$. This optimization surface contains the value function trained using a direct amortized policy. The black diamond, denoting the estimate of this direct policy, is generally near-optimal, but does not match the optimal estimate (red star). Iterative amortized optimizers are capable of generalizing to these surfaces in each case, reaching optimal policy estimates.

The results, averaged across states and random seeds, are shown in Figure 13. CEM (gradient-free) is less efficient than Adam (gradient-based), which is unsurprising, especially considering that Adam effectively approximates higher-order curvature through momentum terms. However, Adam and CEM both require over *an order of magnitude* more iterations to reach comparable performance with iterative amortization. While iterative amortized policy optimization does not always obtain asymptotically optimal estimates, we note that these networks were trained with only 5 iterations, yet continue to improve and remain stable far beyond this limit. Finally, comparing wall clock time for each optimizer, iterative amortization is only roughly $1.25\times$ slower than CEM and $1.15\times$ slower than Adam, making iterative amortization still substantially more efficient.

### B.3    ADDITIONAL 2D OPTIMIZATION PLOTS

In Figure 1, we provided an example of the suboptimal optimization resulting from direct amortization on the `Hopper-v2` environment. We also demonstrated that iterative amortization is capable of automatically generalizing to this optimization surface, outperforming the direct optimizer. To show that this is a general phenomenon, in Figure 14, we present examples of corresponding 2D plots for each of the other environments considered in this paper. As before, we see that direct amortization is near-optimal, but, with the exception of `HalfCheetah-v2`, does not match the optimal estimate. In contrast, iterative amortization is able to find the optimal estimate, again, generalizing the unseen optimization surfaces.

### B.4    ADDITIONAL OPTIMIZATION & THE AMORTIZATION GAP

In Section 4, we compared the performance of direct and iterative amortization, as well as their estimated amortization gaps. In this section, we provide additional results analyzing the relationship between policy optimization and the performance in the actual environment. As we have emphasized previously (see Section 3.2.2), this relationship is complex, as optimizing an inaccurate $Q$-value estimate does not improve task performance.

The amortization gap quantifies the suboptimality in the objective, $\mathcal{J}$, of the policy estimate. As described in Section A.3, we estimate the optimized policy by performing additional gradient-based optimization on the policy distribution parameters (mean and variance). However, when we deploy this optimized policy for evaluation in the actual environment, as shown for direct amortization in Figure 15, we do not observe a noticeable difference in performance. Thus, while amortization may find suboptimal policy estimates, we observe that the actual difference in the objective is either too small or inaccurate to affect performance at test time.

Likewise, in Section 4.2.2, we observed that using additional amortized iterations during evaluation further decreased the amortization gap for iterative amortization. However, when we deploy this

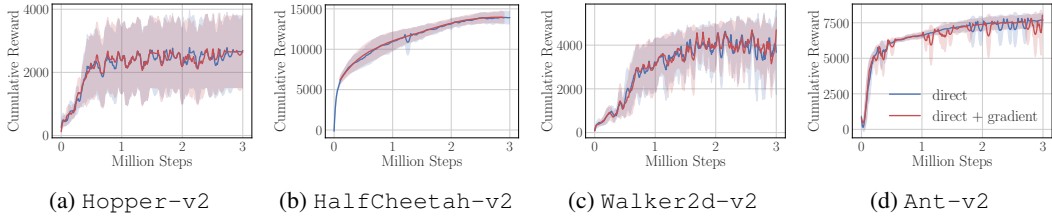

|     |     |     |     |
| --- | --- | --- | --- |
| (a) `Hopper-v2` | (b) `HalfCheetah-v2` | (c) `Walker2d-v2` | (d) `Ant-v2` |

Figure 15: **Test-Time Gradient-Based Optimization**. Each plot compares the performance of direct amortization vs. direct amortization with 50 additional gradient-based policy optimization iterations. Note that this additional optimization is only performed at test time.

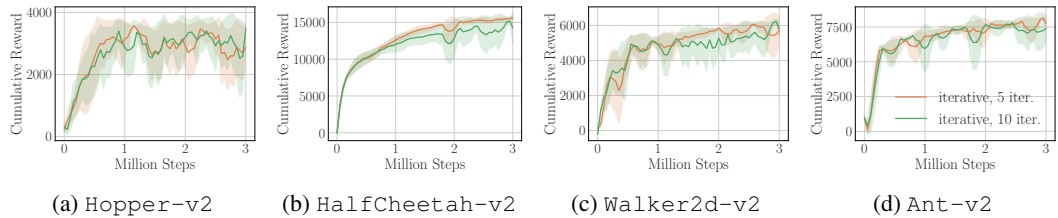

|     |     |     |     |
| --- | --- | --- | --- |
| (a) `Hopper-v2` | (b) `HalfCheetah-v2` | (c) `Walker2d-v2` | (d) `Ant-v2` |

Figure 16: **Additional Amortized Test-Time Iterations**. Each plot compares the performance of iterative amortization (trained with 5 iterations) vs. the same agent with an additional 5 iterations at evaluation. Performance remains similar or slightly worse.

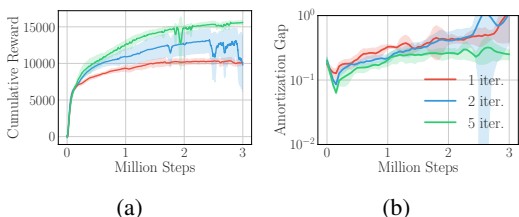

|     |     |
| --- | --- |
| (a) | (b) |

Figure 17: **Iterations During Training**. (a) Performance and (b) estimated amortization gap for varying numbers of policy optimization iterations per step during training on `HalfCheetah-v2`. Increasing the iterations improves performance and decreases the estimated amortization gap.

more fully optimized policy in the environment, as shown in Figure 16, we do not generally observe a corresponding performance improvement. In fact, on `HalfCheetah-v2` and `Walker2d-v2`, we observe a slight *decrease* in performance. This further highlights the fact that additional policy optimization may exploit inaccurate $Q$-value estimates.

However, importantly, in Figures 15 and 16, the additional policy optimization is only performed for evaluation. That is, the data collected with the more fully optimized policy is not used for training and therefore cannot be used to correct the inaccurate value estimates. Thus, while more accurate policy optimization, as quantified by the amortization gap, may not substantially affect evaluation performance, it does play a significant role in improving training.

This was shown in Section 4.2.4, where we observed that training with additional iterative amortized policy optimization iterations, i.e., a more flexible policy optimizer, generally results in improved performance. By using a more accurate (or exploitative) policy for data collection, the agent is able to better evaluate its $Q$-value estimates, which accrues over the course of training. This trend is shown for `HalfCheetah-v2` in Figure 17, where we observed the largest difference in performance across numbers of iterations. We generally observe that increasing the number of iterations during training improves performance and decreases the amortization gap. Interestingly, when performance dips for the agents trained with 2 iterations, there is a corresponding increase in the amortization gap.

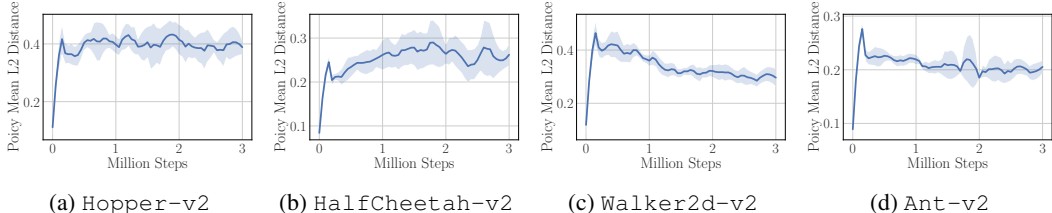

(a) `Hopper-v2`     (b) `HalfCheetah-v2`     (c) `Walker2d-v2`     (d) `Ant-v2`

Figure 18: **Distance Between Policy Means**. Each plot shows the L2 distance between the estimated policy means from two separate policy optimization runs at a given state. Results are averaged over 100 on-policy states at each point in training and over experiment seeds.

## B.5 MULTIPLE POLICY ESTIMATES

As discussed in Section 3.2.1, iterative amortization has the added benefit of potentially obtaining multiple policy distribution estimates, due to stochasticity in the optimization procedure (as well as initialization). In contrast, unless latent variables or normalizing flows are incorporated into the policy, direct amortization is limited to a single policy estimate. To estimate the degree to which iterative amortization obtains multiple policy estimates during training, we perform two separate runs of policy optimization per state and evaluate the L2 distance between the means of these policy estimates (after applying the `tanh`). Note that in MuJoCo action spaces, which are bounded to $[-1, 1]$, the maximum distance is $2\sqrt{|\mathcal{A}|}$, where $|\mathcal{A}|$ is the size of the action space. We average the policy mean distance over 100 states and all experiment seeds, with the results shown in Figure 18. In all environments, we see that the average distance initially increases during training, remaining relatively constant for `Hopper-v2` and `HalfCheetah-v2` and decreasing slightly for `Walker2d-v2` and `Ant-v2`. Note that the distance for direct amortization would be exactly $0$ throughout. This indicates that iterative amortization does indeed obtain multiple policy estimates, maintaining some portion of multi-estimate policies throughout training.

