# OpenReview forum: "Iterative Amortized Policy Optimization"
_ICLR.cc/2021/Conference — Reject_

### Official Review · AnonReviewer4 · 2020-10-26
**OK submission, but confusing and lacking depth**

**Rating:** 6
**Confidence:** 4

**Review:**

Summary
The paper studies the popular soft actor-critic (SAC) algorithm, and points out that it does not find the most optimal actor (action distribution) that maximizes the critic (Q value). The reason is that SAC uses “direct amortization” (a feed-forward network from state to Gaussian params for the action distribution), and hopes that simple gradient descent (eqn (13) in the SAC paper) can sufficiently minimize loss (eqn (10) in SAC) over time, which is not true as Fig 6 in this paper suggests.

To solve this issue, the paper suggests an “iterative amortization” method that optimizes the Gaussian params through a few iterations of 1st order optimization (Alg 2). The optimization is done implicitly with a neuralnet that takes the policy gradient as input (eqn (10)). The resulting algorithm reduces the amortization gap (Fig 6) and improves policy performance over time (Fig 5).

Recommendation
Overall, I’m slightly inclined to accept this paper. The amortization gap is an interesting and often ignored phenomenon. The paper offers a simple solution and performs convincing experimental study. However, the presentation is very confusing, as the focus is selling this A + B solution (A = iterative amortization, B = SAC), rather than analyzing the “amortization gap” issue in full detail. Consequently, the experiments are not comprehensive enough.

Strengths
1. The paper reveals that the SAC policy isn’t always fully optimized. It’s common to assume that gradient descent + large enough network + enough data will eventually optimize the policy. But as Fig 6 suggests, it might not be the case.
2. The paper suggests a simple solution through iterative amortization. This plug-and-play trick outperforms vanilla SAC on two Mujoco tasks (Fig 5).


Weaknesses
1. The paper stops at selling a solution instead of going beyond to analyze the amortization gap. Some natural questions:
(a) There are multiple ways to fully optimize the policy, such as using a larger SGD coeff and running more SGD steps. Do they work?
(b) If the amortization gap is an issue, do you see a problem in the expressivity of Gaussian? Does a more general form of policy (e.g. in the original SQL paper) further reduce the gap and improve performance?
(c) In the case of discrete actions, where the optimal policy can be found directly (simple softmax), does the fully optimized policy outperform direct optimization? Any relevant papers?

2. The presentation is very confusing.
(a) The motivation reads like “Here’s A and here’s B. Let’s try A + B”. If the paper pointed out SAC’s amortization gap earlier and how it seriously impacts performance, the paper would be easier to understand.

(b) Confusing terminology.
- p_{\theta}: Initially described as the agent’s “parametric distribution”, later called “policy prior parameters”, then finally declared a uniform distribution.
- f_{\phi}: Should not use the same symbol for both direct and iterative amortization.
- “policy distribution parameters”: Easily confused with “policy network parameters”. Better called “action distribution parameters”.
- “are policy networks providing fully-optimized policy estimates”: Estimates could mean the Q value. Better called “policy objectives”.

(c) Lack of experimental rigor.
- In Fig 1 and 2, which state and which training iteration is being studied? Why should the reader trust the plots?
- Fig 6 shows that the amortization gap in Ant-v2 does not hurt SAC’s performance. Could you explain why?

Other feedbacks
I put in a lot of effort in understanding this paper and re-summarizing it in this review. I’d hope the authors understand my confusion and restructure the paper correspondingly.

---

> ### Author Response · Authors · 2020-11-14
> **Response to reviewer 4**
>
> Thank you for your review.
>
> The focus of the paper is on introducing improved policy optimization techniques that are both accurate and efficient. We focus on SAC, as it is conceptually simple and sidesteps the potentially confounding aspect of learning an action prior (see e.g., Abdolmaleki et al., 2018). One possible benefit of using a more powerful optimizer is in reducing the amortization gap, however, there are other benefits, such as having policy optimizers that can directly transfer to new value estimates (Fig. 8d) or obtain similar quality estimates faster (Fig. 4c). We are not *selling* iterative amortization + SAC specifically, but this provides a useful setup for isolating and demonstrating this new policy optimization technique.
>
> *Regarding the amortization gap*: a) One previous work did investigate using a more aggressive training procedure for direct amortized networks (Lagging Inference Networks and Posterior Collapse in VAEs, He et al., 2019), which was able to somewhat mitigate the amortization gap. However, because this relies on gradient-based optimization, this now requires another hyperparameter/loop to determine the number of updates before the policy is optimized. Further, because of the reliance on direct amortization, this still cannot be readily transferred to new value estimates. b) A more general form of policy (e.g. the Boltzmann policy used in Abdolmaleki et al., 2018 or SQL) would reduce what is referred to as the *approximation* gap, i.e. the gap between the policy estimate and the posterior policy. However, even in the experiments from these papers, SAC generally performs on-par, suggesting that an entropy-regularized Gaussian is a reasonable approximation. This could also be due to the difficulties of estimating more complex policies, as one is still relying on estimated Q-values. c) In the case of discrete/categorical actions, one can potentially estimate the optimal policy directly. Thus, a separate parametric policy would be unnecessary. If the action space is too large, then, in theory, one could apply iterative amortization or other iterative optimizers to a parametric policy, however, this would require some method for differentiating through discrete actions.
>
> *Presentation*: We apologize that the presentation was unclear. We discuss the amortization gap after the preliminaries in Section 2.3. It is also mentioned in the introduction. Again, we are not specifically advocating using SAC, but it provides a useful setup for isolating policy optimization. If you have suggestions of ways to reorganize the presentation, we would welcome them. We have attempted to present the material in a way that is accessible to those coming from variational inference as well as RL.
>
> *Terminology*:
>
> *The prior*: We presented a general formulation of RL as inference, which includes a prior distribution. This is meant to show that iterative amortization is not limited to SAC. We used consistent notation throughout ($\theta$ for priors, $\phi$ for approximate posterior (policy)).
>
> *Amortized optimizer*: We used the same notation in Eq. 9 and 10 to make it clear that these are both learned mappings. **We have placed comments alongside these equations to make it clear that they are distinct.** Note that these are only used in the initial presentation.
>
> *Policy distribution parameters*: **We have replaced policy distribution parameters with action distribution parameters where appropriate.**
>
> *Policy estimates*: **We have made this change.**

---

> ### Author Response · Authors · 2020-11-14
> **Continued response to reviewer 4**
>
> *2D Plots*: As we discuss in the appendix, the 2D plots are generated from on-policy states for a direct amortized policy (Fig. 1) and an iterative amortized policy (Fig. 2). Both are evaluated at the end of training (3 million steps), on the final collected evaluation episode. To create Figure 1, we initially plotted the objective over two dimensions every 50 steps. Most policy estimates were *near* but not *at* the optimal estimate. The example in Figure 1 is a more substantial gap, although note that the amortization gap is still roughly 1 or less, consistent with our other results. We applied iterative amortization on this state with the value network from the direct policy. The result is shown in the figure. We have *many* other examples of this type of plot, and **we have included examples plots from the other environments in the appendix (Figure B.3)**. To generate Figure 2, we found a state on Ant with a multi-modal objective function along 2 action dimensions, specifically, the 9th state of the episode. We then ran multiple runs of iterative amortization, starting from random policy mean initialization, and plotted 4 runs (2 for each mode). Finally, we note that our analysis code is included in our submission.
>
> *Amortization gap*: We stress that the amortization gap is not *directly* comparable across setups and runs, as they are evaluated with separate value functions and policies.  Further, one cannot *directly* connect the amortization gap with task performance, as the amortization gap is estimated on an objective that includes the policy entropy, as well as an estimated Q-value, which may contain bias (see Fig. 3a). Finally, we note that it is challenging, if not, impossible to exactly evaluate the amortization gap on these environments. We attempt to estimate the gap by running SGD from the amortized policy’s estimate. However, this could potentially exploit the learned Q-value estimator, finding a policy that is estimated as better but not actually better; we observed this when initializing SGD from a standard Normal policy. Initializing from the amortized estimate helps to restrict SGD to the local region of policy-space. While we acknowledge that this analysis is limited, we feel that it is nevertheless an important step toward quantifying the quality of policy optimizers. To clarify these points, **we have provided a more in-depth discussion and analysis of the amortization gap in the main paper as well as the appendix.**
>
> We attempted to present iterative amortization through a lens that will permit those familiar with RL and variational inference to understand. If you have other specific ideas on ways to improve the presentation, we would welcome your comments.

---

### Official Review · AnonReviewer1 · 2020-10-28
**A learning-to-learn style proposal for policy optimization**

**Rating:** 5
**Confidence:** 2

**Review:**

### Summary

The paper proposes to replace the actor/policy network with an iterative version to encode the action distribution parameters, which is inspired by prior work on iterative amortized optimization. This scheme generates the action distribution parameters for each state at the end of an inner loop which takes the objective gradient wrt to the parameters also as input (as opposed to a regular actor network which does one forward pass). This seems similar to prior work on learning to learn/meta learning but the presentation can be improved (more detailed comments below).  Experimental evidence is given for benefits over SAC in mujoco environments.

Overall, the general idea seems interesting and useful, but the novelty is unclear given the prior work; the clarity of the presentation can be improved which makes an accurate assessment of the significance harder.

### Detailed Comments

* In the introduction, authors motivate the idea of amortization by making a distinction between optimizing the distribution versus a network whose parameters output the distribution. Can the authors provide an example of an algorithm that does the former for clarity (i.e. an example of a non-amortized policy -- what would that look like hypothetically?)

* The agent's policy is denoted by $p_\theta$ (prior to Eq (1)) and then with $\pi$ later. Comment below Eq (8) makes a reference to $\theta$ as prior parameters. This presentation is unclear and needs additional information to disambiguate.

* What do the dependencies in Figure 1 right denote? For instance, $p_\theta$ has a dependence on $p_{env}$ in this figure, but it's not clear what that means.

* How is $\lambda$ initialized in the inner update loop? The capability to find multiple modes seems dependent upon providing diverse initializations, which seems unspecified.

* The value overestimation section discussion seems to suggest that the iterative scheme does. worse than the direct even with changing $\beta$. Not clear what the take away from this section is in the context of the rest of the paper.

---

> ### Author Response · Authors · 2020-11-14
> **Response to reviewer 1**
>
> Thank you for your review. We hope that we can clear up any confusion.
>
> Regarding your detailed comments:
>
> *Non-amortized policies*: As discussed in the related work section (2.4), previous works have applied optimization methods like CEM to optimize policy distribution parameters. Purely by convention, this is commonly applied in model-based control, however, we note that QT-Opt applies CEM to model-free control. Non-amortized policies are typically less common in RL, as this generally requires a many iterations per time step. We show this in Figure 4c, where a tuned Adam and CEM baseline require hundreds of iterations to optimize the policy distribution, whereas iterative amortization is near-optimal in a handful of iterations. To be clear, in the optimizer setups, the mean and variance of the policy are treated as free parameters, which are optimized to maximize the estimate of $\mathcal{J}$.
>
> *Prior vs. policy*: We presented a more general formulation, with both a prior ($p_\theta$) and approximate posterior ($\pi_\phi$), as iterative amortization is not restricted to the SAC setup. This distinction is standard in previous work, see, e.g., maximum posteriori policy optimization (Abdolmaleki et al, 2018). $\theta$ corresponds to the action prior’s parameters, which we leave as uniform in this paper, and $\phi$ corresponds to the parameters of the amortized policy. These parameters are entirely distinct throughout the paper.
>
> *Dependencies*: The dependencies denote computational dependencies arising from the probabilistic computation graph. Each arrow receives a *sample* of the input variable, e.g., $ \mathbf{s}_t \sim p ( \mathbf{s}_t | \cdot)$. The output of the arrow is a distribution (circle) or a Q-value (diamond). Solid arrows denote action prior and Q-value calculation, whereas dashed arrows denote policy optimization computations. Specifically for $p_\theta$, the arrow denotes the conditional dependence in $p_\theta (\mathbf{a}_t | \mathbf{s}_t)$, which could be parameterized by a neural network (Abdolmaleki et al., 2018) or could be uniform (Haarnoja et al., 2018).
>
> $\lambda$ *Initialization*: The policy distribution initialization is a design choice, however, it is typically initialized from the prior (Marino et al., 2018ab). In this paper, because the prior is uninformative, we initialize from a standard Normal distribution. **We have clarified this further in the appendix A.4.** Indeed, initialization can affect the degree of exploration, however, this is also governed by stochasticity in the gradient estimates due to action sampling. As shown by Greff et al., 2019, this is already sufficient for recovering multiple modes.
>
> *Value Overestimation*: The takeaway from the value overestimation is *not* that iterative performs worse than direct amortized optimization. Rather, it shows that naively applying previous techniques (taking the minimum of two Q-networks) does not work well, as the flexibility of iterative amortization to move around the policy landscape allows it to exploit regions of uncertainty. This is an important point in applying more powerful optimizers to policy optimization, which we feel will be a helpful consideration for future work in this area.
>
> Regarding novelty, we would like to reiterate that this paper is, to the best of our knowledge, the first to explicitly connect KL/entropy-regularized policy networks with the concept of amortization. This naturally raises questions about the quality of the policy, i.e. whether it actually optimizes the objective, which relates to the amortization gap. To improve policy optimization, we apply iterative amortization, which has not been demonstrated previously in RL. We show that this results in a policy optimization scheme that *1)* outperforms both gradient-based and gradient-free optimizers in terms of efficiency (Fig. 4c), *2)* outperforms direct amortization in terms of task performance (Fig. 5 & Fig. 8a), and *3)* generalizes to unseen value estimators (Fig. 8d). These are useful contributions for the RL community, opening an analysis of policy optimizer quality, improving upon standard policy optimization techniques, and paving the way for more powerful/efficient policy optimization techniques by highlighting accompanying issues with value overestimation.

---

> > ### Comment · AnonReviewer1 · 2020-11-21
> > **Thanks for the clarifications**
> >
> > Thank you for the clarifications.  I have another question:
> >
> > A key input to the procedure is $\nabla_\lambda \mathcal{J}$, where $\mathcal{J}$, the regularized objective is an expectation over trajectories. For the gradient, I presume we have to condition on a given state, so there is still an expectation over actions and reward samples (generated from the combination of the prior policy params and the environment). How is this sample estimate of the gradient defined within Algorithm 2? (i.e. the third argument to the function $f_\phi$).
> >
> > In the paper, you comment briefly saying, "...*this scheme requires some way of estimating J online, e.g. through
> > a parameterized Q-value network (Mnih et al., 2013) or a differentiable model (Heess et al., 2015)*..."  More precisely, do you take an empirical mean of Monte carlo samples for a in the approximation to $\mathcal{J}$ and then differentiate through this?

---

### Official Review · AnonReviewer2 · 2020-10-28
**Simple method for improving maxent policy optimization**

**Rating:** 5
**Confidence:** 3

**Review:**

**Contributions**: The authors propose to use iterative amortization for policy optimization to help reduce suboptimality in policy optimization. They find that like in variational inference for generative modeling, iterative amortization is able cover multiple modes of distributions and lower the amortization gap, and show slightly improved performance on mujoco gym benchmarks. Additionally, they find that the iteratively amortized policy is able to better exploit the learned Q function and lead to more overestimation bias, and address this by tuning a parameter to make the Q-function backups more pessimistic to compensate.

**Questions and Concerns**: It is not entirely clear to me where the better performance observed from using iterative amortization derives from in the benchmark results. For example, in Hopper and HalfCheetah, we see that the iterative amortization slightly outperforms the direct baseline, yet the the difference in amortization gaps seems negligible throughout training. On the other hand, in the Ant environment, the amortization gap of the direct method is much higher, yet the learning curves of both methods are practically identical. One potential issue is that the objective of the policy update depends on the *learned* Q-function instead of the true returns from the environment, and so a better inference procedure may not even necessarily be a benefit. For this reason, we often consider schemes like trust regions for policy updates that explicitly limit policy optimization.

Some additional questions:
To what extent is the amortization gap alleviated by direct amortization but larger, more expressive networks? Does doing so provide similar improvements in performance?

In Figure 3b, it appears that with the iterative schemes, the new policy consistently differs the old one (what exactly is pi_old in this case?) by a significant amount. Should we be concerned here that the learned policy (and presumably also the Q-functions) are not really coming close to converging? This further suggests to me strictly trying to have the policy better approximate this noisy oscillating Q may not actually be beneficial behavior.

**Summary**: The authors demonstrate that iterative amortization enables some improvement to policy optimization in a maximum entropy RL algorithm. However, it is not entirely well understood to me how precisely iterative amortization leads to such benefits. Regarding novelty, the presented method is a straightforward application of a technique from variational inference to maxent RL, so is perhaps a bit incremental in that sense. Overall, I like the method since it provides a very simple way to improve learning by plugging in iterative amortization, but think it needs a to provide a better understanding of why it helps.

---

> ### Author Response · Authors · 2020-11-14
> **Response to reviewer 2**
>
> Thank you for your review.
>
> Regarding amortization gaps, we stress in the paper than it is not possible to *directly* compare amortization gaps across different optimizers/runs. This is because the amortization gap depends on the value function, which itself depends on the policy optimizer for future value estimates. Further, amortization gaps are estimated using the learned value function; thus, a large amortization gap could also indicate that the optimizer successfully avoids regions of value overestimation, which may be found by SGD during gap estimation. While we attempted to counteract this phenomenon (see the appendix A.3), we admit that this estimation procedure is limited. Likewise, the amortization gap does not *directly* translate into a performance gap, as it is the gap in the regularized objective, $J$, which includes both the estimated discounted environment return as well as the entropy. The amortization gap is only an indicator that the policy optimizer is more fully optimizing the value estimator, which is beneficial for collecting data during training. **We have added a more in-depth discussion and analysis to the main paper and the appendix.**
>
> Regarding direct policy network size, in a separate set of experiments, we have observed that the width of a direct policy network does not substantially affect the resulting performance on MuJoCo tasks. For instance, with SAC, we observed that a policy of width 32 does not perform noticeably worse than the default width of 256. This suggests that the suboptimality of direct amortization is truly due to the restricted *functional* form rather than the particular architecture.
>
> This relates to a key point that we emphasized in the paper: direct amortization only receives policy updates *after* outputting the distribution estimate, whereas iterative amortization uses feedback *during* estimation to update the distribution. The important aspect is that direct amortization requires multiple interactions to find optimal policies, while iterative amortization (through policy gradients) can instantly generalize to new value estimates (Fig. 8d).
>
> The larger KL divergence in the policy is not necessarily a negative aspect; in fact, it can be helpful for adapting to new value estimates. This KL corresponds to a KL between the current policy ($\pi_\textrm{new}$) and the policy from 1,000 steps before ($\pi_\textrm{old}$). We note that this is estimated using the action samples from eval episodes, with only a single run of policy optimization per state. Thus, due to this sampling-based KL estimate and the multiple modes potentially captured by iterative amortization, we should inherently expect iterative amortization to have a larger KL than direct amortization. Indeed, **we have added an additional section to the appendix demonstrating that iterative amortization maintains multiple policy estimates throughout training.** We note that while the policy KL remains larger, the performance (Fig. 5) remains stable, and Q-values converge and do not oscillate.
>
> Finally, we would like to emphasize that although this paper may at first appear to be a “straightforward application” of iterative amortization to RL, identifying this connection and understanding/mitigating the accompanying issues in value overestimation is non-trivial. We see the fact that the technique seems obvious (in hindsight) as a *strength* of the paper. We put considerable effort into making the paper accessible to a wide audience. Identifying and applying techniques across disparate areas is not only useful, but also helps to bring along previously developed insights, e.g., amortization, suboptimality, normalizing flows, etc. We feel that our paper makes important contributions toward connecting these concepts with RL.

---

> > ### Comment · AnonReviewer2 · 2020-11-20
> > **Overall opinion remains unchanged**
> >
> > I have read the author's response and the updated paper, and my overall rating of the paper remains unchanged.
> >
> > My main concern remains that it has not been clearly demonstrated that amortization gaps themselves are a major problem in maximum entropy reinforcement learning. Given the results that increasing policy network size (which should decrease the amortization gap without affecting the approximation gap) does not seem to affect performance, this suggests to me that amortization gap is really not the issue. One caveat with direct amortization here is that perhaps just increasing network expressivity is not sufficient for reducing amortization gap, and more training of the policy relative to the Q-functions is needed as well (whereas iterative amortization can adapt to the changing Q-function on the fly). However, if the suboptimality of direct amortization is due to restricted functional form as the authors suggest, then I do not believe this paper should be focusing on amortization gaps, but instead focus more on the *approximation gap*.
> >
> > Given that iterative amortization is able effectively capture a more expressive policy, it is entirely possible that the benefits we see are actually derived from reducing the approximation gap rather than the amortization gap. It would be thus important to compare against other methods that incorporate more expressive policies like normalizing flows, as well as expanding the discussion of related work in this direction. For example, https://arxiv.org/abs/1905.06893 demonstrate normalizing flows are able to improve performance with SAC and perform better exploration.
> >
> > Minor additional suggestion for related work: https://arxiv.org/abs/1911.01831 directly parameterize advantages with a normalizing flow that they  can use directly sample from to use as a policy, thus removing the amortization gap entirely.

---

> > > ### Author Response · Authors · 2020-11-21
> > > **Clearing up misunderstandings**
> > >
> > > There appears to be some misunderstanding here, which we would like to clear up. The amortization gap is purely due to the quality of the *optimizer*, whereas the approximation gap is purely due to the flexibility of the *distribution*. Thus, in our paper, direct and iterative amortization have exactly the same approximation gap (for a given value function), as they both use tanh-transformed Gaussian distributions. The *only* difference is in the amortization gap. Iterative amortization, by definition, is a strictly more powerful optimizer class. Indeed, in our instantiation, direct amortization (a fixed mapping) is a special case of iterative amortization (an iterative mapping).
> > >
> > > While one might initially expect that increasing the network size for direct amortization would reduce the amortization gap, this is *not* the case for precisely the reason you gave. Deficiencies in optimization performance are not (necessarily) due to the flexibility of the network architecture; they are a result of not being sufficiently trained to optimize the objective. After value updates, a direct amortized policy must update its state-action mappings for all relevant states. This an *inherent* functional limitation of direct amortization. In contrast, iterative amortization can *immediately* optimize new value estimates by relying on gradient-based feedback during optimization. We have demonstrated multiple instances of this capability in the paper and have shown that it is useful.
> > >
> > > We cited several initial works that used normalizing flows to parameterize policy distributions. **Thank you** for these additional references, which we will incorporate into the paper. However, this seems to be a misunderstanding. We are *not* proposing a new policy distribution, but rather an improved method for performing policy optimization. While we agree that exploring the approximation gap in RL is a useful direction, this is not the focus of our work. One insight does apply to both the approximation and amortization gaps, though: improving the policy is useful (for exploration) during data collection.
> > >
> > > To close, we would like to re-state that iterative amortization, by its very nature, is a more powerful optimization technique than direct amortization. The fact that iterative amortization performs as well or better than direct amortization (with the same policy distribution class) across environments clearly demonstrates that the policy optimizer *does* matter. Note that this can *only* impact the amortization gap; the approximation is the same in both cases. As shown in our analysis, as well as in previous works, improving the policy is helpful for improving data collection (*exploration*), rather than merely improving the current performance (*exploitation*). While we have attempted to quantify the amortization gaps in each case, we have provided clear caution in over-interpreting the results, as it is difficult to exactly measure and compare gaps. Nevertheless, we feel that this direction is a useful contribution. Finally, the fact that we are even discussing amortization vs. approximation gaps for policy optimization already demonstrates the beneficial contributions of this paper in bringing these concepts to RL.

---

> > > > ### Comment · AnonReviewer2 · 2020-11-21
> > > > **Updated comment**
> > > >
> > > > Apologies to the authors, in my previous comment I misspoke regarding approximation gaps, and I will try to clarify my comments here.
> > > >
> > > > When I mentioned approximation gaps, I was really only concerned about the benefits of multimodal policies for exploration, and not necessarily in how it contributes to more optimal inference. As the authors mention in the paper, iterative optimization is able to sample from *multiple modes* across different runs, which can lead to benefits in exploration. However, this multimodality in exploration can also be captured by using a more flexible policy distribution like normalizing flows, and which have been shown to also improve performance. One potential explanation for the improved performance from iterative optimization could be that it is just due to this multimodality and better exploration, hence why I believe comparisons to methods utilizing flows or other more flexible policy classes would be important.
> > > >
> > > > If it were indeed true that amortization gaps were a major issue, then we should also see benefits with direct amortization provided we have an expressive enough policy network and ran a lot more optimization the policy relative to the value function.
> > > > However, I again emphasize that quite a few prior works actually try to *limit* how fast the policy changes (for example using trust region constraints, Fujimoto et al in the TD3 paper also proposed performing *fewer* policy updates compared to the Q-function), suggesting that reducing suboptimality in policy optimization is not necessarily beneficial.
> > > > While it is certainly an advantage of iterative amortization that it can adapt to changing targets much more easily, it is unclear how much this actually benefits in the setting of the benchmark results given that our algorithmic choices force the Q-function to change fairly slowly for stability reasons (for example slowing down changes in the target values of the Bellman backup with target networks and target policies).
> > > >
> > > > It's also unclear to me if this adaptability is really useful in other RL settings either. Is there a scenario in which we ever need to suddenly switch between two very different value estimators? If trying to incorporating online feedback during a trajectory, it seems like a lot of work would already need to be done to update the value function anyways, so simply updating with direct amortization doesn't seem too bad anyways.

---

### Official Review · AnonReviewer3 · 2020-10-29
**Interesting work and connection on iterative amortized optimizers**

**Rating:** 5
**Confidence:** 3

**Review:**

This paper draws an interesting connection to variational inference, categorizing current policy optimization methods with KL regularization as direct amortized optimizers. The paper shows how direct amortized policy optimization can be suboptimal and proposes a new class of method called iterative amortized policy optimization. This ‘iterative’ amortized policy optimization performs iterative optimization before the regular optimization step, offering several advantages: better at reaching the optima, better at sampling multiple modes, and support more rapidly changing policy. The paper shows nice visualization to support these claims and also run benchmark experiments to show its improvement.

This paper has done a good job introducing ‘iterative amortized policy optimization’, providing support with both good visualizations at a simpler level (to showcase what it is actually doing) and empirical evaluation on complex benchmarks. I think the contribution is novel and introduces a different type of optimization that is shown to improve current policy optimization methods.
Despite the paper being an overall good paper, I think the experiments should be much better supported and recommend a borderline reject unless the issues below can be addressed.

The results look convincing overall but I think more runs should be conducted. Experiments on benchmarks only conduct 4 runs. There is high variance in results due to random seeds especially in Mujoco domains. Henderson et al. 2018 (https://arxiv.org/pdf/1709.06560.pdf) show that two different groups of runs with 5 random seeds each can lead to significant difference in performance on the same algorithm. It is hard to say that iterative amortized optimization helped improve performance with only 4 runs.

Furthermore, it is not mentioned at all in the main paper that for HalfCheetah and Walker2D, iterative vs direct use different model architectures (it is mentioned in Appendix A.4 Q-value section).

These two environments in particular were environments that showed significant difference between the two methods, and makes the reader wonder whether this improvement was due to optimization method difference or architectural change.
I think at least a justification in this particular design choice is needed or perhaps all results for both models can be shown to help readers understand the general performance.

Lastly, it would also be helpful to mention how many runs were conducted for model-based value estimate experiment in Figure 8a.

Other minor details:
I thought it would be helpful to mention where experiment details can be found in the appendix for each figures or sections in the main paper.

At the bottom of p.4 last paragraph of section 3.1, should \nabla_\phi J be \nabla_\lambda J?

---

> ### Author Response · Authors · 2020-11-14
> **Response to reviewer 3**
>
> Thank you for your review.
>
> We agree that using a sufficient number of seeds is important for ensuring reproducible results. **We have run an additional seed for each environment in the model-free setting**, shown in the updated draft of the paper. Iterative amortization still consistently outperforms direct amortization. **We are in the process of running further seeds as well for all settings.**
>
> Regarding the value network architecture, we found that a larger value network architecture (closer to that used in Maximum a Posteriori Policy Optimization (Abdolmaleki et al., 2018)) was necessary to obtain the reported SAC results on HalfCheetah with our code base. Thus, to be fair, we treated the value architecture as an additional hyperparameter, reporting the results from the best performing value network in each case. **We have included these results in the updated appendix for full transparency.**
>
> For model-based experiments, we also ran 4 seeds. **We are currently running additional seeds for each setup**, although, again, we do not claim that these are state-of-the-art model-based results, but rather a proof-of-concept.
>
> **We have attempted to better connect the main paper to specific sections in the appendix.**
>
> In Section 3.1, this is meant to be $\nabla_\phi J$. This refers to the gradients w.r.t. the iterative optimizers parameters, just as in the case of direct amortized networks (or direct encoders in VAEs).

---

> > ### Comment · AnonReviewer3 · 2020-11-23
> > **Thank you for the update**
> >
> > Thanks for doing more runs.
> >
> > Also, I've checked the appendix, and only one seed is used to compare the four combinations of direct/iterative, A/B architecture (Figure A.2). How did you compare and determine the best performing value networks for each case? Could you include all runs you did when comparing them? Comparing performance with only one run does not show anything because of variance.
> >
> > Even though these experiments are not supposed to be state-of-the-art and proof-of-concept I think the results should be reliable enough to assert that the difference is due to the using different optimizer rather than by chance.
> >
> > Minor comments: Section 4.2.2 still says 4 runs instead of 5 runs.

---

### Author Response · Authors · 2020-11-14
**Response to all**

We would like to thank all of the reviewers for their comments. We have incorporated these comments into an updated draft and supplementary, and we have **bolded** these changes in our individual responses. We ask that the reviewers take this into consideration when making their final assessment. Here, we address general comments, with individual comments to follow.

**Novelty**: We would like to emphasize that although this paper may at first appear to be a “straightforward application” of iterative amortization to RL, identifying this connection and understanding/mitigating the accompanying issues in value overestimation is non-trivial. Policy optimization performance is rarely (if at all) considered in continuous control; direct policy networks are the default choice. To the best of our knowledge, we have provided the first paper to explicitly evaluate this assumption, as well as the first paper to explicitly link policy networks with amortization. Further, we provide a *new*, iterative method for performing policy optimization, demonstrating the power and generalization of this method in both model-free and model-based settings. This opens *new* opportunities toward efficiently performing policy optimization in more complex and unseen settings, as we have shown in the paper.

**Analysis**: Part of the motivation for introducing iterative amortized policy optimization is to obtain improved estimates of the policy distribution, as quantified by the amortization gap. However, there are other benefits to iterative amortization, such as obtaining multiple estimates and generalizing to new value functions, both of which we have shown. As we stress in the paper, the amortization gap is only *indirectly* related to the performance in the environment, as it depends on the KL regularizer and the agent’s Q-value estimate. Further, the gap itself is small in comparison to the objective, and as we demonstrate in our updated analysis, closing this gap does not noticeably improve the *evaluation* performance. Rather, an improved policy optimizer is helpful for improving *training*, as demonstrated in Figure 7, allowing the agent to better obtain state-action pairs in the environment with high estimated value. We have also provided a new section in the appendix analyzing the degree to which iterative amortization obtains multiple policy estimates throughout training.

We will close by reiterating that we have introduced a new technique for policy optimization, providing a new framing of conventional approaches to policy optimization through the lens of amortization. Our experiments provide clear demonstrations of the efficacy and efficiency of iterative amortized policy optimization in a variety of settings, and our analyses provide new directions toward evaluating policy optimization techniques. Finally, by connecting amortization with policy optimization, we are hopeful that this paper will inspire the community to bring further insights from variational inference to reinforcement learning.

---

### Decision · Program_Chairs · 2021-01-07
**Final Decision**

**Decision:**

Reject

**Comment:**

I think there is a lot to commend in this paper: the general approach for training f_phi in this way is creative and interesting, the discussion of the amortization gap is thought-provoking, and the general idea is not something that I have seen in the literature before. That said, the reviewers raise a number of important concerns about the approach, chief of which is that the paper's explanation for why the method works is questionable. The proposed method can be viewed as simply a more expressive policy architecture. In fact, I suspect strongly that the modest increase in performance is likely explained by this alone. The discussion with Reviewer 2 in particular makes this issue very clear. I don't think the authors offered a very compelling response to this. Therefore, I think there are just too many question marks about this approach to accept the paper for publication at this time.

I do however think that this line of research is very promising, and I would encourage the authors to continue this work and flesh out the evaluation to be more rigorous and complete, understand whether the increase in performance comes down simply to increased expressivity or if the discussion of amortization is closer to reality, and also address other concerns raised by R2 and the other reviewers.